# Single Cell RNA Sequencing: A New Frontier in Pancreatic Ductal Adenocarcinoma

**DOI:** 10.3390/cancers14194589

**Published:** 2022-09-22

**Authors:** Maroun Bou Zerdan, Malek Shatila, Dhruv Sarwal, Youssef Bouferraa, Morgan Bou Zerdan, Sabine Allam, Merima Ramovic, Stephen Graziano

**Affiliations:** 1Department of Internal Medicine, SUNY Upstate Medical University, Syracuse, NY 13210, USA; 2Department of Gastroenterology, Hepatology & Nutrition, The University of Texas MD Anderson Cancer Center, Houston, TX 77030, USA; 3Department of Internal Medicine, Cleveland Clinic Foundation, Cleveland, OH 44118, USA; 4Faculty of Medicine, University of Balamand, Beirut 0000, Lebanon; 5Department of Hematology and Oncology, SUNY Upstate Medical University, Syracuse, NY 13210, USA

**Keywords:** pancreatic ductal adenocarcinoma, single cell RNA sequencing, fibroblasts, transcriptomes

## Abstract

**Simple Summary:**

Pancreatic cancer has a very low survival rate for several reasons. One of those is primarily due to the difficulty in diagnosing it at an early stage. For this reason, it is important to refine our understanding of this disease to guide the development of new diagnostic and therapeutic modalities to combat this fatal illness. Here we attempt to provide a review of current progress in utilizing single-cell RNA sequencing (scRNA-seq) techniques in the molecular profiling of pancreatic ductal adenocarcinoma.

**Abstract:**

Pancreatic ductal adenocarcinoma is a malignancy with a high mortality rate. It exhibits significant heterogeneity in metabolic pathways which are associated with its progression. In this review, we discuss the role of single cell RNA sequencing in unraveling the metabolic and clinical features of these highly malignant tumors.

## 1. Introduction

Pancreatic cancer is a highly lethal malignancy of the gastrointestinal tract. Despite its rarity, it has one of the highest mortality rates among GI cancers, with a five-year survival of 11.5% overall [1,2]. It is the seventh leading cause of cancer mortality in the world [3,4] owing to its highly aggressive nature [5], and the annual incidence is expected to increase in the coming years [6]. Tumors may arise in either the endocrine or the exocrine pancreas, depending on the cell of origin. Patients with cancers of the endocrine pancreas have a better prognosis, and survive about two years longer than those with exocrine tumors [7]. Pancreatic ductal adenocarcinoma (PDAC) is the predominant type of exocrine tumor and will be the focus of this review. These comprise more than 90% of all pancreatic cancers [8] and have a five-year survival rate of about 11%, which is among the lowest of all cancers [1,2]. This dismal survival rate is attributed primarily to difficulty in diagnosis at a localized stage and the lack of effective treatment modalities. Pancreatic cancer is usually asymptomatic in its early stages, and by the time patients develop symptoms it has usually progressed to an advanced stage [9]. Moreover, early screening in the general population is not feasible given its relatively low incidence and the lack of reliable diagnostic biomarkers at present [10]. Only 10–20% of patients are eligible for surgical resection at presentation [11,12], with those undergoing surgery being prone to relapse [13]. Patients receiving chemotherapy often show limited responses and usually end up succumbing to disease progression [13]. Gemcitabine-based and FOLFIRINOX regimens are the mainstays of pharmacotherapy depending on tumor stage and resectability [14]. Given its high mortality rate and resistance to treatment, pancreatic cancer is anticipated to be the second leading cause of cancer deaths globally by 2030 [15]. For this reason, it is imperative to refine our understanding of this disease to guide the development of new diagnostic and therapeutic modalities to combat this fatal illness. Here we attempt to provide a review of current progress in utilizing single-cell RNA sequencing (scRNA-seq) techniques in the molecular profiling of pancreatic ductal adenocarcinoma.

## 2. Discussion

There is a well-established genetic component to pancreatic cancer and PDAC [16,17,18]. Past genomic studies have consistently shown mutations in four major genes (*KRAS*, *TP53*, *CDKN2A*, *SMAD4*), with a constellation of other molecular aberrations that are of uncertain significance. These findings suggest that PDAC most often originates from a disruption of common signaling pathways downstream of the above mutations. While a growing body of evidence supports this, it alone does not explain the range of clinical characteristics, in terms of aggressiveness and treatment resistance, seen with these tumors [13]. Given the continuous division and proliferation of tumor cells, numerous biological and genetic differences arise within these tumor cells, creating a complex intra-tumoral heterogeneity (ITH).

Additionally, tumor tissue differentiates into distinct cell types depending on microenvironmental factors, adding to the diversity within these tumors [19]. Heterogeneity has been shown to be a major driving force of drug resistance in cancer [20]. Genetic mutations alone do not suffice in explaining tumor pathogenesis—rather, we must also factor in the tumor microenvironment (TME) and epigenetic mechanisms of tumoral evolution [21]. Next-generation whole-exome sequencing has played a critical role in recent years, uncovering novel driver mutations and signaling pathways involved in different forms of the disease [17,22] and identifying similar gene signatures in primary and metastatic lesions [23]. These studies were essential in the early steps of individually tailored chemotherapy for PDAC [13], a subject we will explore later in this review. Despite this, genomic studies can only go so far. A known obstacle to genome sequencing is low tumor cellularity [24]. PDAC lesions typically exhibit a strong desmoplastic reaction [25] and are notorious for their low cellularity, with a typical PDAC lesion being composed of only 5–20% neoplastic cells [26]; this lowers the quality of the data obtained. Moreover, exome-sequencing of a cohort of PDAC survivors found no difference in somatic mutations that could explain the improved tumor course in these rare patients compared to the majority, suggesting that looking at genetic mutations alone may not be enough to study this disease [27]. More importantly, genetic techniques cannot give enough details on the nature of the tumor cell clusters and the microenvironment to be a helpful tool in understanding tumor heterogeneity [27,28].

Transcriptomic techniques use RNA to identify which genes are actively expressed within a cell population. First, RNA must be isolated from a sample population of cells and then broken down into smaller fragments. Reverse transcriptases are then added to form DNA from the isolate and amplified through a polymerase chain reaction. The generated strands are then spread out on a matrix, and fluorescent probes are added sequentially to identify the base sequence in each obtained DNA strand. Once this library has been created, the data must then be analyzed to quantify the transcriptional activity of each gene, allowing us to explore tumor heterogeneity and microevolution [29,30]. One of the earliest applications of this technique in PDAC was in 2011 by Collison et al.: they used bulk-sequencing to identify three PDAC subtypes, each showing the expression of unique sets of genes [31]. They described a classical subtype expressing mainly epithelial and adhesion-associated genes, a quasi-mesenchymal type with a high expression of mesenchyme-associated genes, and an exocrine-like subtype that expressed digestive enzyme genes [31]. The first-ever classification of PDAC into subtypes with significant implications for the prognosis and treatment of the disease was made possible through the aforementioned study [13]. This classification has been updated and improved many times over the years. It established for the first time that different microenvironmental signatures are associated with different clinical outcomes based on the tumor cell subtype [13]. That said, bulk-based approaches produce composite data that reflects the average gene expression across a cell [32,33], which lacks the granularity to resolve the complexity of the PDAC microenvironment and inevitably ends up missing out on a large fraction of data [13,34]. Techniques that can operate at a higher resolution, ideally at the level of an individual cell, would be more suited for that purpose.

The advent of scRNA-seq techniques provided a much-needed solution to the above problems. With scRNA-seq, researchers could study the tumor heterogeneity and microenvironment at an unprecedented resolution [21], potentially revolutionizing our understanding of PDAC. Peng et al. demonstrated this by outlining a more comprehensive taxonomy of pancreatic cancer. Through scRNA-seq, the researchers found two ductal cell clusters and at least seven subclusters with unique gene upregulations in each [35]. They also identified five sub-clusters of T-lymphocytes and macrophages, each comprising unique immune ecosystems. While Peng’s seminal study showed immense promise for applying scRNA-seq to pancreatic cancer, it is limited by its failure to include other stromal and immune cells in its analysis of the tumor microenvironment [13]. It is imperative to consider the various non-cancer cell populations when exploring these tumor microenvironments [36]. In particular, immune cells need to be scrutinized, as patient survival has been associated with immune infiltration, with significant correlations to immune cell types and inflammation signatures present [37,38].

Nonetheless, this study proved the translational use of scRNA-seq in PDAC. ITH and TME are major barriers to cancer therapy and contribute strongly to drug resistance [13]. Tiriac et al. showed that transcriptomic signatures could predict the effectiveness of different chemotherapy options in a given patient [39]. With time, scRNA-seq could very likely usher in a new age of precision medicine in pancreatic cancer research and therapeutics.

Our review aims to explore the implications of scRNA sequencing technologies on the future of PDAC. First, we will explore how scRNA techniques have refined our understanding of the PDAC subtypes by looking at the cell clusters inherent to each. We will review how transcriptomic techniques can be potentially used to identify tumor sensitivity to different chemotherapy regimens and guide future treatment guidelines. Finally, we will take a better look at the PDAC microenvironment and its relation to tumor properties.

### 2.1. Genomic Approaches to Pancreatic Cancer Profiling

Recent forays into pancreatic cancer genomic profiling highlight a need to focus on overall tumor genomics rather than discrete mutations that may only be present in a minor subset of all tumors [16]. The earliest genomic studies on this subject were limited in the technologies and the methodologies available, and thus their scope was restricted to identifying key mutations common to PDAC. Four primary genes were recurrently implicated in the disease—*KRAS*, *TP53*, *CDKN2A*, *SMAD4*—and future genomic studies served to add to the growing list of mutations common to it [13]. Jones et al. were able to establish 12 regulatory pathways (apoptosis, DNA damage control, cell cycle checkpoint regulation, hedgehog signaling, cell adhesion, integrin signaling, c-Jun signaling, *KRAS* signaling, invasion, GTPase-dependent signaling, TGF-B signaling, and Wnt/notch signaling) affected by the genetic mutations in PDAC that brought a “rudimentary understanding to a very complex disease” [13,16]. The researchers also found that while all 12 pathways were commonly dysregulated in all tumor samples, the specific gene mutations that led to that disruption varied largely between tumors [13]. This means that we are still uncertain about the origin of these tumors, and genomic studies have been hard-pressed to provide sufficient answers. Over the years, genomic studies were only able to confirm previously discovered genetic mutations, with the main recent addition to the genomic literature being the implication of axon guidance genes (semaphorins, slits)—traditionally regulating normal neuronal migration and positioning—in tumor growth, survival, invasion, and angiogenesis [40]. So far, all genomic studies painted the picture of a genetically homogenous disease, which however behaves in a very heterogeneous fashion clinically. Genomics has been unable to help distinguish between different tumors and explain their unique clinical course. Waddel et al. identified chromosomal structural variation as an important mechanism of DNA damage in PDAC, and thus attempted to use it as grounds to classify different PDAC subtypes [17]. The researchers were able to develop four groups depending on chromosomal stability. Subtype 1 was the stable subtype with little variations. Subtype 2 consisted of significant focal events on few chromosomes, three mutations scattered across the tumor chromosome, and four consisted of unstable chromosome structures exhibiting many structural variation events. While this classification was the first of its kind, it had a minimal bearing on clinical practice.

### 2.2. Transcriptomic Approaches to Pancreatic Cancer Profiling

Genetic mutations alone cannot inform us of what is happening at the molecular level; instead, gene expression would provide a more accurate and detailed account of how PDAC tumors develop and progress [27]. RNA-sequencing technologies have therefore become critical in appreciating PDAC pathogenesis and the complex tumor heterogeneity and microenvironment underlying it. The first accurate attempt to classify PDAC came from Collison et al. [31]. Using bulk sequencing on a sample of 27 previously micro-dissected tumors, the authors were able to identify three unique gene signatures that were differentially expressed in the samples, suggesting at least three different variations of PDAC. The first form of PDAC identified was the classical subtype expressing high amounts of adhesion-associated and epithelial genes. Classic PDACs were more dependent than other subtypes on *KRAS* mutations, and GATA binding protein 6 (GATA6) was predominantly expressed by classical PDACs and found to have a subtype of a specific function. The following form described was the quasi-mesenchymal (QM-PDA) subtype which, owing to its name, highly expressed mesenchyme genes and expressed little to no GATA6.

The final subtype is the exocrine-like expressing digestive enzyme genes, which have since been considered an artifact of contamination by normal pancreatic tissue surrounding the studied samples [41]. This classification of PDAC came with direct implications on prognosis and treatment outcomes, meaning that, for the first time, identifying tumor subtype had a genuine relevance to clinical practice. In this case, tumors of the classical subtype typically had a better survival profile. They responded better to erlotinib, while QM-PDA tumors showed worse survival and a better response to gemcitabine [11]. This hallmark study introduced an entirely new paradigm in the conceptualization of pancreatic cancer and opened the floodgates for RNA-based research. The first update to Collison et al.’s classification system came from Moffit et al. in 2015 [41]. Previously, the study looked at gene expression only across tumor cells. To reduce confounding from the gene expression of neighboring tissue, Moffit et al. included the analysis of normal tissue and stroma in their study, incidentally uncovering unique gene expression from the cancer stroma in the process [41]. A significant update on the previous model for was observed for several reasons. First, they showed that the exocrine-like PDAC subtype was very likely normal tissue that was included accidentally, which was also later demonstrated by Puleo et al. [42]. Second, they found that the QM-PDA subtype’s expression of mesenchymal genes likely arose from the inclusion of stromal factors—by identifying stromal gene expression, Moffit et al. were able to redefine the second PDAC subtype as basal-like [42]. This left us with an even more precise idea of pathogenesis: PDAC tumors were of either classical or basal-like origin, with two different possible stromal activation patterns. The activated stroma was associated with macrophage-related and fibroblast activation protein genes and correlated with worse median survival times given their involvement in disease progression. This new model saw a cumulative effect of tumor and stromal subtypes on prognosis, which added another layer to our understanding of PDAC and shed some light on the important interplay within tumors that needs to be taken into [42].

One year later, Bailey et al. published another transcriptomic study describing yet another classification scheme with four tumor subtypes [43]. They added to Collison’s work by identifying novel transcription networks implicated in Collison’s taxonomy, renaming them to better reflect the molecular pathology, and introducing a new type: immunogenic PDAC. QM-PDA became squamous PDAC and was also found to express TP63 instead of being GATA6 highly. The classic subtype was the pancreatic progenitor subtype, defined by the expression of apomucins PDX1 and FOXA2/A3 genes, all involved in pancreatic endoderm cell-fate determination and pancreas development. Finally, the immunogenic subtype shared much in common with the pancreatic progenitor class, with the addition of significant immune infiltrate and expression of genes associated with nine different immune cell types and phenotypes. Moffit’s basal-like subgroup had no apparent niche in this classification scheme and seemed to be a mix of different Bailey/Collison subtypes. This discrepancy in describing PDAC may arise from a difference in the quality of the data used and the various research methodology.

Years later, Moffit’s binary classification was reproduced by The Cancer Genome Atlas consortium (led by Raphael BJ) [18] and later by Puleo et al. [43]. Inspired by Moffit’s analysis of the tumor-stromal component, the latter decided to take a closer look at the tumor micro-environment. They identified four stromal signatures in PDAC: a vascular stroma composed of fibroblast and endothelial cells, an activated stroma of extracellular matrix-associated gene expression, an inflammatory stroma with high levels of IL-6 and macrophage activation, and, finally, an immune stroma with the increased expression of immune response genes. They conclude that neoplastic cells could either be basal-like or classical, which comprises the heterogeneity of the tumor. In contrast, the tumor entity could be pure classic, immune classic, pure basal-like, or a hybrid with both classical and basal-like cells (showing either a desmoplastic or a stroma-activated reaction). Though it would come to be improved in the future, this classification serves a pivotal role in drawing attention to the extensive role the tumor microenvironment has in PDAC pathogenesis and incorporating that microenvironment into the classification. Moreover, it demonstrated the limits of what bulk sequencing can tell us about tumor heterogeneity. Recent work by Raphael et al. shows that molecular classification of pancreatic cancer is highly influenced by stromal cell infiltration, suggesting that in the absence of adequate cell purification, results from bulk sequencing studies are often tainted by adjacent non-tumor cell signals, an issue easily avoidable with single cell technologies [18].

### 2.3. scRNA-seq Provides New Insight into Pancreatic Cancer

Bernard et al. carried out one of the first scRNA-seq investigations into pancreatic cancer to identify the microenvironmental and neoplastic cell transcriptional changes associated with tumor progression [44]. Their primary findings were that rare cells with malignant potential were present in pre-cancerous lesions likely responsible for neoplastic progression. Moreover, they found a dynamic shift in the tumor microenvironment over time, with the pre-cancerous lesions exhibiting high concentrations of CD4^+^ and CD8^+^ lymphocytes and special cancer-associated fibroblasts (CAFs) called myofibroblasts (myCAFs). Figure 1 depicts the role of the tumor microenvironment in cancer stem cells enrichment in PDAC.

Adenocarcinomas on the other hand showed deficient levels of the lymphocytes mentioned above and high levels of immune fibroblasts (iCAFs) that facilitated a pro-tumoral milieu for invasion and metastasis [44]. This work highlighted the power of scRNA-seq techniques and strongly endorsed its use in studying this complex disease as its higher resolution picked up data that would have certainly been lost with bulk sequencing [13]. Peng J et al. further demonstrated the power of single-cell techniques by studying the transcriptomic profiles of around 42,000 neoplastic and controlling pancreatic cells [34]. To start with, they identified two unique ductal cell clusters, types 1 and 2. Type 2 ductal cells had a high expression of previously reported poor prognosis markers. At the same time, cancer-related genes were not detected in control samples, suggesting that these type 2 cells are the primary source of the malignancy. Type 1 cells, on the other hand, expressed genes related to normal pancreatic function and some disease-related features, suggesting at least some level of ordinary functioning. Type 1 cells were divided into two subcategories, while type 2 cells were grouped into seven subclusters after further analysis, each distinguished by a specific gene signature. Importantly, while subcluster 3 was found in virtually all patients, subclusters 1, 2, 4, 5, and 6 were only exclusive to some patients, demonstrating tumor heterogeneity. Importantly, the first three subclusters of ductal type 2 cells strongly represented the precursor lesion to PDAC, pancreatic intraepithelial neoplasia (PanIN), indicating a stage-specific progression pattern [34]. In addition to this, Peng J et al. identified 24 subsets of immune cells (5, 6, 5, and 8 subgroups of T-cells, B-cells, macrophages, and fibroblasts, respectively). This creates a far more intricate and detailed concept of PDAC micro-composition and helps refine the existing two-clade classification system. More importantly, it once again reflected the potential of single-cell analysis by allowing us to evaluate the entire spectrum of dynamic molecular programs for cell lines, which is more representative of reality as opposed to defining artificially discrete categories of cells. Although an extraordinary undertaking, Peng J et al. (2019) received some criticism for not looking more closely at other stromal/immune elements, and CAFs were not discussed despite their significant contribution to shaping the PDAC microenvironment, as discussed earlier [13,44].

### 2.4. Tumor Heterogeneity Analysis Using scRNA-seq

Xu et al. have shown the utility of scRNA-seq in characterizing tumor heterogeneity in liver metastases from PDAC [45]. Their analysis identified multiple lineages of cancer cells within metastatic lesions, including five different types of ductal cells, and temporal trajectory mapping for cell lineage evolution was performed to elucidate the origins of cells within the metastatic lesions under investigation, highlighting the role that scRNA-seq has in predicting the composition of metastatic lesions or primary PDAC when tissue is only available from one source, as well as investigating the driver mutations behind neoplastic progression and dissemination. Lee et al. [46] demonstrated agreement of scRNA-seq guided tumoral heterogeneity analysis with that from surgically resected specimens, providing further evidence for utilizing scRNA-seq in guiding therapeutic targeting and response prediction in metastatic PDAC.

### 2.5. Predictive Modeling Using scRNA-seq

Gopalan et al. [47] have explored the predilection for origination of PDAC in primed precursors using scRNA-seq to analyze a specific sub-population of acinar cells, which they termed “edge cells”. These cells demonstrated markers of regression and de-differentiation that could, at least partly, explain the origins of PDAC. Their transcriptomes were shown to lie closer to malignant PDAC cells than normal acinar cells on the transformation continuum. Their results also showed the presence of these “edge cells” in the setting of chronic pancreatitis, a condition that has been previously shown to be associated with a higher risk of developing PDAC, as well as being associated with *KRAS* mutations, which are known to be one of the most common mutations seen in PDAC tissue.

scRNA-seq has also been used to predict tumor behavior and differentiation and the resultant impact on metastasis and prognosis in PDAC. Tang et al. performed an analysis of scRNA-seq data to delineate tumor composition with respect to immune cell profiles and were able to identify higher numbers of Tregs and macrophages as well as lower numbers of dendritic cells in metastatic PDAC as compared with localized forms of the disease [48]. Through scRNA-seq analysis of the prevalent immune cells and associated tumor cells, their work focused on *HMGB3* as a hub gene in the metastatic transition of PDAC. This work highlights the utility of scRNA-seq in identifying the mechanisms and driver genes underlying the transition from localized to disseminated disease in PDAC. Chen et al. utilized a similar analysis on data from the Genome Sequence Archive (GSA) to identify tumoral sub-populations and variations in their characteristics, such as copy-number variations and the degree of epithelial-to-mesenchymal transition [49]. Their study was, however, limited by sample size, and thus they were unable to use scRNA-seq data to generate their prognostic and diagnostic models, though they successfully used scRNA seq data for identifying the differential expression of genes.

Ren et al. demonstrated the use of scRNA-seq in identifying the genetic evolution of cancer stem cells to invasive ductal cells in PDAC and the impact of this change on tumor progression [50], with their model highlighting five genes that were associated with significant prognostic impacts.

Fang et al. [51] highlighted the metabolic pathway differentiation in single tumor cells that demonstrated higher glycolytic activity and greater malignant potential. Knockout models from the potentiated sub-populations of tumor cells showed decreased proliferation rates, highlighting both the importance of metabolomics in guiding cancer therapeutics in PDAC as well as the importance of scRNA-seq in elucidating these metabolic profiles.

### 2.6. Stromal Biology and the Role of Cancer-Associated Fibroblasts

PDAC is characterized by abundant desmoplasia that takes up to 90% of the total tumor volume and contains a mixture of cells, including CAFs [25]. CAFs may stimulate cancer progression by secreting extracellular matrix (ECM) and are thought to be derived from resident mesenchymal cells: pancreatic stellate cells. However, while they were traditionally thought to play a tumor-supportive role in PDAC pathogenesis exclusively, this view has recently been challenged [52]. Ohlund et al. were the first to investigate these fibroblasts using scRNA technology and identify two distinct clusters of CAFs, the same ones reproduced by Bernard et al. a year later [52]. myCAFs, nonneoplastic contractile cells expressing αSMA, were involved in stroma remodeling and formed a peri-glandular ring surrounding cancer cell clusters. iCAFs, on the other hand, were located distant from the neoplastic cells and were the result of paracrine activation of pancreatic stellate cells, primarily by IL-6. Biffi et al. expanded on this by showing that pancreatic cancer cells could secrete a large amount of TGF-B and IL-1 [53]. While TGF-B facilitated a juxtacrine reaction that strongly activated myCAF signaling and suppressing the induction of iCAF populations, IL-1 activated JAK/STAT signaling distant to the tumor, promoting the growth of an iCAF phenotype. This endorses the idea of a very dynamic tumor microenvironment, as stromal elements seem to be interconvertible rather than terminally differentiated. In 2019, Elyada et al. uncovered a new class of CAF called antigen-presenting fibroblasts (apCAFs) that uniquely expressed Major Histocompatibility Complex (MHC) II genes. While their role was uncertain, and Dominguez et al. later argued against the presence of apCAFs, they nonetheless suggested that there was still a lot more to be learned about PDAC neoplastic and stromal cell lines while also showcasing some of the difficulties inherent in single-cell methods [54].

Other stromal components have also been studied using scRNA-seq. Wang et al. highlighted the differences in tumor-associated polymorphonuclear cells using an scRNA-seq analysis [55]. Their work delineated four distinct tumor-associated neutrophil lineages with different metabolic and immune behaviors and identified a key regulator driving this differentiation. The glycolytic upregulation mediated by this regulator imparted a pro-tumor effect and portended a poorer prognosis in these tumors, providing support to the utilization of single cell analysis to support therapeutic selection and prognostication.

### 2.7. Current Classification of Pancreatic Cancer

Chan-Seng-Yue et al. have combined genomic, bulk-RNA, and snRNA technologies to propose a comprehensive classification of PDAC cell lineages [56]. This study describes five categories of PDAC cells: Classical-A/B, Basal-A/B, and a hybrid subtype with signatures from both. It is proposed that classical tumors follow a default molecular pathway in tumor pathogenesis, as they are the most abundant cell type and display a strong upregulation of transcription factors commonly expressed in normal ductal cells such as GATA6. On the other hand, Basal-like cells are acquired in a subset of tumors and are likely an indicator of tumor progression. This is supported by the fact that classical tumors were more frequently found in early-stage cancers, while basal-like tumors were more likely present in later stages. Most tumors, at single-cell resolution, consist of both sub-types; however, the varying proportion of each cell cluster creates a transcriptional continuum the authors refer to as hybrid tumors. While there has yet to be a definitive taxonomy for pancreatic cancer, the attempts to create one are numerous and ongoing [57]. Almost all published articles exploring molecular subtypes describe some iteration of the classification taxonomy scheme proposed by Collisson, Bailey, and Moffit [58,59]. By growing consensus, two distinct subtypes of PDAC exist: the basal/squamous subtype, and the classical/pancreatic progenitor subtype [60,61].

The body of research discussed here does well in showing the complexity of pancreatic cancer, but even then, it only scratches the surface. For instance, Chan-Seng-Yue et al. [48] show that sub-type switching can occur as tumors progress in response to therapy. Molecular classification is thus a far more dynamic continuum rather than a static system. Considering the shortcomings of genomic and bulk transcriptomic studies that are often plagued by difficulties in obtaining samples, microenvironmental signals confusing results, and the comparatively low quality of data obtained, single cell technologies hold a lot of promise in unraveling the intricacies of pancreatic cancer. They have already revolutionized our understanding of this deadly disease [62,63]. scRNA has the potential to play an instrumental role in translational research in the years to come, and is already in use to explore therapeutic options in different PDAC cell lines [64].

### 2.8. Status of PDAC Chemotherapy

The current guidelines for the treatment of pancreatic cancer involve a limited group of chemotherapy regimens that alternate depending on various factors such as disease stage, patient tolerance, and regimen effectiveness. These regimens are either gemcitabine-based (as monotherapy or in combination), or a mix of fluorouracil, leucovorin, irinotecan, and oxaliplatin (FOLFIRINOX) [14,65,66]. Which treatment is administered depends strongly on the current state of the patient’s disease. Patients with resectable PDAC can be started on neoadjuvant therapy with either gemcitabine + S-1 therapy or gemcitabine monotherapy in patients with poor tolerance to S-1 [65]. Adjuvant therapy with gemcitabine (as monotherapy, or with capecitabine or nab-paclitaxel if tolerable) is recommended post-resection as soon as the patient can tolerate all cycles [44,61]. Chemoradiotherapy in this case has been shown to confer little to no survival benefit over chemotherapy alone and is mainly beneficial in unresectable, locally advanced diseases to control pain [65].

For this reason, we will be limiting the scope of our discussion to chemotherapy only. In cases of locally advanced disease, gemcitabine or FOLFIRINOX are both recommended first-line treatments [65], with FOLFIRINOX particularly recommended for fit patients considering its improvement of disease-free and overall survival [65]. As for metastatic disease, FOLFIRINOX is strongly recommended as the first-line treatment, replaced by gemcitabine-combination therapies only if patients cannot tolerate or did not benefit from the former regimen [14,65,66]. Unfortunately, despite these existing medications, treatment of pancreatic cancer remains very challenging, with five-year survival rates as low as 3% and 8% in the UK and the US, respectively [67].

### 2.9. Mechanisms of Drug Resistance in Pancreatic Cancer

The high mortality of PDAC stems from the very nature of the disease, which leads to a resistance to conventional chemotherapy regimens [68]. PDAC is characterized by a strongly desmoplastic stroma which reduces drug penetration into the area [69]. Moreover, PDAC cells employ strong immune-evasion mechanisms that reduce the effectiveness of the patients’ immunity and drugs targeting the immune system [70]. However, the most important mechanism of drug resistance is the extensive tumor heterogeneity present in PDAC [31,33,34,41,43,57]. While there is a body of literature on the genetic landscape of PDAC development and progression, it has been challenging to translate that into effective therapy for afflicted individuals because of the multiple compensatory survival mechanisms exhibited by PDAC cells. Given the variety of genetic mutations, the diversity of pathways affected means that even if highly dysregulated pathways are targeted with treatments, other pathways may be utilized by cancer cells to ensure their survival [70]. That said, our knowledge of tumor genetics can help inform clinical practice. Whole-exome sequencing of PDAC typically finds genomic lesions that are hypothetically actionable in 50% of cases and may even result in a change in clinical management in up to 30% of [64]. Some examples of this include using Larotrectinib in patients with an identified TRK gene fusion, gemcitabine/cisplatin regimens in patients with BRCA1/2 or PALB2 mutations, and pembrolizumab monotherapy in cases with high microsatellite instability [64]. As discussed earlier, an understanding of tumor genetics alone will not suffice in tackling this disease, rather, we must take a closer look at the tumor microenvironment and epigenetic factors. These mechanisms can be summarized in Figure 2.

### 2.10. Utility of Molecular Profiling in Predicting Treatment Outcomes

In one of the earliest studies taking a more holistic look at pancreatic cancer, Collison et al. aimed to test whether the phenotypic differences in their derived subtypes corresponded with a differential response to treatment [31]. Their study subjected human derived PDAC cell lines to either gemcitabine or erlotinib and assessed their sensitivity to each. They found that the classical subtype dependent on mutant *KRAS* mutations was susceptible to erlotinib, while the quasi-mesenchymal subtype was partial to gemcitabine. This appeared to be a breakthrough in PDAC treatment, with the theoretical possibility of exploiting such subtype-specificity to develop targeted therapies [31]. Bailey et al., for instance, encouraged the clinical trial testing of immunomodulators in treating their newly discovered immunogenic PDAC subtype 2. Peng J et al. found that CDK1 inhibitors significantly suppressed the propagation of a specific line of proliferative ductal cells [37]. Altogether, these findings support the use of molecular profiling in the clinical setting. However, it is essential to bear in mind that the real-world application of molecular profiling is not as straightforward as the described research may suggest. Pancreatic cancer is far too complex to be reduced to overarching subtypes, where a unique therapy regimen exists for each type. Chan-Seng-Yue et al. describe two cases where patients start off with one class of pancreatic cancer and were discovered to have a completely different subtype after disease progression or chemotherapy in a phenomenon they described as subtype switch [57].

Moreover, each patient’s disease is an amalgamation of different cancer, immune, and fibroblast cell lines, coming together to form a highly unique molecular profile [71]. This makes a one-size-fits-all approach to treating pancreatic cancer virtually impossible. Instead, it becomes imperative to create highly personalized treatment schemes for these patients, targeted therapies homing in on the specific molecular mechanisms driving tumor progression in each case. To this end, various researchers have called for the implementation of molecular subtyping using transcriptomic techniques—scRNA-seq in particular—in clinical trials [72,73,74]. Others still recommend profiling the tumor tissue of all patients coming in with PDAC [75] with the recent National Comprehensive Cancer Network (NCCN) guidelines for PDAC strongly recommending molecular tumor analysis [64].

### 2.11. Pharmaco-Typing Patient-Derived Organoids to Develop Treatment Modalities

Organoids are a 3D cell culture derived directly from the individual’s cells, the tumor cells. They allow us to mimic the in vivo characteristics of PDAC, including its development, pathogenesis, genomic and transcriptomic factors, and drug response [76]. With the advent of co-culturing techniques, organoids can even allow us to elucidate microenvironmental factors that influence carcinogenesis [76,77,78]. Furthermore, they conveniently overcome the two biggest problems molecular profiling faces as they can be generated from minimal amounts of tissue; a single EUS biopsy would be enough [79]—and this can be done in a short period [76]. Organoids are thus suitable for developing experimental models to guide personalized treatment and genetic research of PDAC patients [80]. One widespread use for organoids in clinical research involves a process known as “pharmaco-typing”. They can be subjected to different medications, with the resulting treatment response data assimilated into already gathered genomic, transcriptomic, or proteomic databases [39]. This approach can identify the figurative Achilles heel of each tumor by discovering which drug compounds are effective while also correlating this response with particular molecular subtypes [70]. Importantly, these assays can be conducted in a short period before even starting clinical therapy, with screens being finished within three to four weeks of receiving a biopsy [80].

Herve Tiriac and David Tuveson have extensively studied organoids in pancreatic cancer, and more recently used organoids to assess the effectiveness of the five most commonly used chemotherapeutic agents on different tumor samples [39,81]. They divided their patient-derived organoids (PDOs) into three groups for each chemotherapy agent: least responsive (highest tertile on the dose-response curve, most resistant), most responsive (lowest tertile, most sensitive), and intermediate (middle tertile). They found that treatment response could be mapped to specific transcriptomic signatures, which could then be used to predict treatment response in other patients with similar signatures [39]. These findings were reproduced by Seppälä et al. with three of the five chemotherapy regimens used by Tiriac et al. To date, these are the two primary studies to utilize PDOs to explore treatment options in pancreatic cancer [82]. Furthermore, they provided sufficient evidence to reasonably predict that PDO may serve as a platform for novel drug discoveries and personalized, precision medicine in pancreatic cancer [83]. While these authors refrain from proposing the implementation of PDO pharmaco-typing as it exists in the clinical setting, they advocate for the validation of its use in PDAC to facilitate such implementation, and indeed, there are several ongoing clinical trials employing PDAC PDOs that aim to do just that [83]. Molecular profiling combined with organoid pharmaco-typing may be the key to conquering the beast that is pancreatic cancer. At the center of such methodologies lies scRNA-seq. scRNA is a critical tool that allows for translating molecular subtypes and drug response curves into clinically meaningful information. It is a tool that will become essential for screening, profiling, and treating patients, and marks the beginning of an exciting era of single-cell genomics in cancer research [84].

However, an important limitation of organoids is the “black-box” nature of organoid cellular adaptation to the culture environment. Whether such adaptation leads to changes in the transcriptome and how the gene expression profile is affected has yet to be elucidated. Thus, cells within these organoids may not necessarily be representative of the proliferating cells in the actual tumoral biopsy specimen.

### 2.12. Limitations of Molecular Profiling Techniques

It is one thing to advocate for molecular techniques to enhance our understanding of pancreatic cancer; whether that is practical is a separate discussion. Molecular profiling techniques are criticized for lack of translatability stemming from several key issues. To begin with, there is great difficulty in obtaining samples for analysis, both because of the anatomically difficult location of the tumor paired with the desmoplastic histology of pancreatic cancer, resulting in very low cellularity [85,86,87,88,89]. Another concern with bringing molecular subtyping to clinical practice is the turnaround time. Pancreatic cancer is a highly aggressive disease notorious for rapid clinical deterioration in most patients. This leads to the question of whether molecular analyses can be carried out quickly enough to be of any value. Finally, creating an efficient system for analysis would require a herculean effort on an organizational level to implement specialized multidisciplinary teams and invest in the bioinformatics tools needed [85]. Together, these factors pose a significant challenge to the use of molecular techniques and must be addressed before a transition to precision medicine in the realm of pancreatic cancer is possible. Various researchers have attempted to demonstrate the feasibility of using molecular profiling in the clinical setting. Dreyer et al. conducted molecular profiling of 95 PDAC samples obtained from endoscopic ultrasound (EUS) biopsies and accurately segregated patients into clinically relevant subtypes, showing that sufficient samples can be obtained [90]. Other studies showed that molecular analyses can be conducted in clinically relevant timelines (<35 days), while still identifying predictive transcriptional features that allow for improved treatment selection [91,92].

Moreover, molecular technologies continue to improve to overcome whatever challenges arise, with novel approaches showing promise in stratifying patient response to treatment in many tumor types [72]. As for pancreatic cancer, several groups have developed informatics designs that purport to identify and stratify patients into clinically actionable subtypes [74] accurately; these have yet to be assessed in clinical trials. Nonetheless, these numerous studies have shown the feasibility of incorporating molecular analysis into current clinical practice to screen patients and assign them to matched chemotherapy treatments, at least within the United States [72]. However, more important than this is the simple fact that it works. Pishvaian et al. conducted a retrospective analysis of 677 PDAC patients who underwent molecular testing of their illness [57]. Actionable alterations were identified in 189 cases, and the overall survival and median overall survival of different patient groups were measured. They found that patients with actionable alterations who received a targeted therapy had significantly longer median overall survival than patients who received unmatched treatments (2.58 years vs. 1.51; *p* = 0.0004), and a longer overall survival than patients without an actionable target (2.58 years vs 1.32; *p* < 0.0001). Interestingly, median overall survival did not differ between patients without an actionable alteration and patients receiving unmatched therapy (*p* = 0.1). These results suggest that molecular profiling has a bearing on clinical practice and could potentially guide our choice in treatment. However, clinical trials are necessary before any new treatment guidelines can be set, and patient-derived organoids may be the perfect tool to bridge this gap [70].

## 3. Conclusions

scRNA-seq analyzes transcriptomic information at the individual cell level. It has been used to discover new cell subtypes, reveal cell heterogeneity, and monitor the dynamic process of disease development, among other things [93]. Compared to traditional RNA-sequencing, scRNA-seq allows insight into the metabolic heterogeneity of malignant cells and provides insight into the operant metabolic pathways within the tumor microenvironment. When applied to clinical samples across tumor types, single-cell sequencing technologies can facilitate an exponential increase in knowledge of the molecular pathways involved in PDAC genesis, local progression, as well as distant metastasis, and provide novel biological targets at the cellular level for improving our approach to the treatment of PDAC.

## Figures and Tables

**Figure 1 cancers-14-04589-f001:**
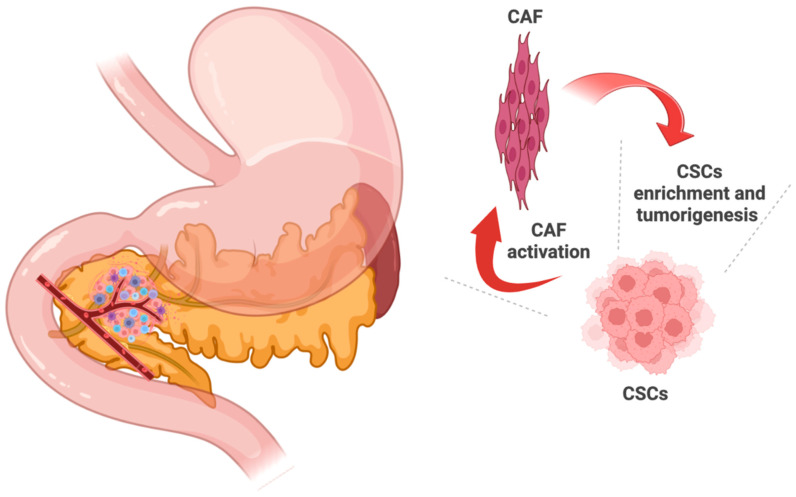
The tumor microenvironment is comprised of a diverse range of immune cells, the extracellular matrix, blood vessels, and cancer-associated fibroblasts (CAFs), and these factors are involved in the aggressiveness of pancreatic cancer cells. CAF: cancer associated fibroblasts. CSCs: cancer stem cells.

**Figure 2 cancers-14-04589-f002:**
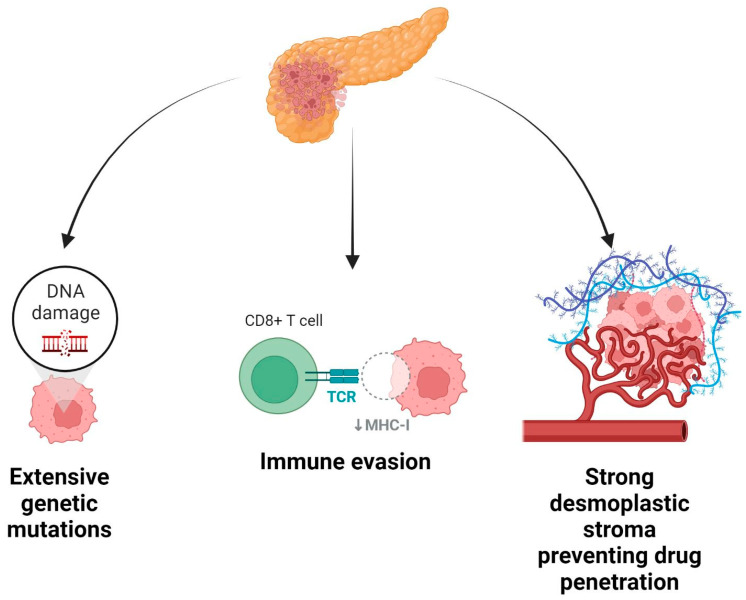
Mechanisms of drug resistance in pancreatic cancer.

## Data Availability

Not applicable.

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
