# Peer review of "Single Cell RNA Sequencing: A New Frontier in Pancreatic Ductal Adenocarcinoma"

_cancers, 2022, doi:10.3390/cancers14194589_

Round 1

Reviewer 1 Report

The authors reviewed the single cell RNA sequencing in pancreatic ductal adenocarcinoma. While the most relevant papers have been included, the structural concept of the manuscript is set up in a way that some of the information is repeated, which breaks the flow of the whole text. For example, high aggressiveness of PDAC is commented on in the Introduction and again at the beginning of the paragraph on Genomic approaches to pancreatic cancer (lines 125 onwards). Profiling data from Peng et al are discussed twice (first on lines 102 onwards, then again lines 244 onwards – the appropriate place for it), then again in lines 377-378.

The content of the review is first introduced in the Discussion part, rather than at the end of the Introduction, and some work on the scRNA-seq has already been talked about beforehands. Study of Chang-Seng-Yue (wrongly spelled as Wue) is described under Current classification of pancreatic cancer (lines 294 onwards), then in the follow up paragraph again.

Some sentences are not clear, for example: line 262-264and lines 355-356 (‘Given the variety in genetic mutations, the diversity of pathways affected has it.’), then lines 425-426.  

‘ScRNA-seq provides new insight into pancreatic cancer’ is one of the smallest chapters, considering that it should have been the focus of the whole review.

Limitations of molecular profiling techniques paragraph would have been better to include towards the end of the manuscript.

The worst issue, however, seen across the entire manuscript, is almost continuous confusion with referencing, which greatly hampered its reviewing. A large number of references are cited in the wrong place, and some are completely wrong (for example ref 49 is a reference on hormone-refractory prostate cancer and ref 82 is on parasite Eimeria).

 The Conclusion paragraph is poorly written.

Author Response

All comments have been addressed. Changes are tracked. Sorry for the delayed response. 

Reviewer 2 Report

This is a well-written and rahter informative review on PDAC but the content does not closely align with the title because the analysis of the sc-RNA Seq approaches to study PDAC is not the major part of the review and the authors have relied too much on other reviews for this. As a result the manuscript looks already outdated.

The review would benefit greatly if the authors were to discuss the contributions of scRNA Seq studies in subtopics of PDAC such as predictive models (eg 10.3389/fgene.2022.892177, DOI: 10.1093/hmg/ddab343. DOI: 10.1080/21655979.2021.1962484), tumour heterogeneity (eg doi: 10.1038/s41420-021-00663-1 doi: 10.1158/1078-0432.CCR-20-3925), putative progenitor cells (eg doi: 10.3389/fcell.2022.798165, doi: 10.1158/0008-5472.CAN-21-0427) and stromal biology (eg 10.1136/gutjnl-2021-326070). Thus the authors should significantly update their manuscript and properly discuss and integrate these findings before the article becomes a review worth publishing.

Patient specific organoids are also discussed as an additional tool for individualised treatments. This is appropriate but an important caveat that needs to be discussed is that the extent to which cells adapt to the culture environment is unknown. This adaptation will be associated with changes in the transcriptome and therefore the gene expression profile is not necessarily the same as that of the proliferating cells in the biopsy.

Author Response

Point by point:

We thank the reviewers for their comments and have addressed them in the revised manuscript.

  1. While the most relevant papers have been included, the structural concept of the manuscript is set up in a way that some of the information is repeated, which breaks the flow of the whole text. For example, high aggressiveness of PDAC is commented on in the Introduction and again at the beginning of the paragraph on Genomic approaches to pancreatic cancer (lines 125 onwards). Profiling data from Peng et al are discussed twice (first on lines 102 onwards, then again lines 244 onwards – the appropriate place for it), then again in lines 377-378.
    1. We have revised the flow of the manuscript for clarity and succinctness.
  2. The content of the review is first introduced in the Discussion part, rather than at the end of the Introduction, and some work on the scRNA-seq has already been talked about beforehands. Study of Chang-Seng-Yue (wrongly spelled as Wue) is described under Current classification of pancreatic cancer (lines 294 onwards), then in the follow up paragraph again.
    1. We apologize for misspelling the cited author’s name and have corrected this.
    2. We have incorporated the suggestion to introduce the review topic at the end of the Introduction section.
  3. Some sentences are not clear, for example: line 262-264and lines 355-356 (‘Given the variety in genetic mutations, the diversity of pathways affected has it.’), then lines 425-426.
    1. Revised as directed.
  4. ‘ScRNA-seq provides new insight into pancreatic cancer’ is one of the smallest chapters, considering that it should have been the focus of the whole review.
    1. We have expanded this section in the revised manuscript.
  5. Limitations of molecular profiling techniques paragraph would have been better to include towards the end of the manuscript. 
    1. We have moved this section closer to the end of the Discussion section.
  6. The worst issue, however, seen across the entire manuscript, is almost continuous confusion with referencing, which greatly hampered its reviewing. Many references are cited in the wrong place, and some are completely wrong (for example ref 49 is a reference on hormone-refractory prostate cancer and ref 82 is on parasite Eimeria).
    1.  
  7. The review would benefit greatly if the authors were to discuss the contributions of scRNA Seq studies in subtopics of PDAC
    1. such as predictive models (eg 3389/fgene.2022.892177, DOI: 10.1093/hmg/ddab343. DOI: 10.1080/21655979.2021.1962484),
    2. tumour heterogeneity (eg doi: 10.1038/s41420-021-00663-1 doi: 10.1158/1078-0432.CCR-20-3925),
    3. putative progenitor cells (eg doi: 10.3389/fcell.2022.798165, doi: 10.1158/0008-5472.CAN-21-0427)
    4. stromal biology (eg 10.1136/gutjnl-2021-326070).

Thus the authors should significantly update their manuscript and properly discuss and integrate these findings before the article becomes a review worth publishing.

We have incorporated the suggested references in the respective sections of the manuscript with the respective citations.

  1. Patient specific organoids are also discussed as an additional tool for individualised treatments. This is appropriate but an important caveat that needs to be discussed is that the extent to which cells adapt to the culture environment is unknown. This adaptation will be associated with changes in the transcriptome and therefore the gene expression profile is not necessarily the same as that of the proliferating cells in the biopsy.
    1. We appreciate this input and now discuss this point in the section for patient-specific organoids.
  2. The Conclusion paragraph is poorly written. —> it has been slightly changed. 

Round 2

Reviewer 2 Report

Ready after spell check